# Identification of Withaferin A as a Potential Candidate for Anti-Cancer Therapy in Non-Small Cell Lung Cancer

**DOI:** 10.3390/cancers11071003

**Published:** 2019-07-17

**Authors:** Jade H.-M. Hsu, Peter M.-H. Chang, Tai-Shan Cheng, Yu-Lun Kuo, Alexander T.-H. Wu, Thu-Ha Tran, Yun-Hsuan Yang, Jing-Ming Chen, Yu-Chen Tsai, Yeh-Shiu Chu, Tse- Hung Huang, Chi-Ying F. Huang, Jin-Mei Lai

**Affiliations:** 1Department of Biotechnology and Laboratory Science in Medicine, National Yang-Ming University, Taipei 112, Taiwan; 2Division of Medical Oncology, Department of Oncology, Taipei Veterans General Hospital, Taipei 112, Taiwan; 3Faculty of Medicine, National Yang-Ming University, Taipei 112, Taiwan; 4Institute of Biopharmaceutical Sciences, National Yang-Ming University, Taipei 112, Taiwan; 5Department of Computer Science and Information Engineering, National Taiwan University, Taipei 106, Taiwan; 6The Ph.D. Program for Translational Medicine, College of Medical Science and Technology, Taipei Medical University, Taipei 110, Taiwan; 7Taiwan International Graduate Program in Molecular Medicine, National Yang-Ming University and Academia Sinica, Taipei 112, Taiwan; 8The Ph.D. Program in Pharmaceutical Biotechnology, College of Medicine, Fu Jen Catholic University, New Taipei City 242, Taiwan; 9Graduate Institute of Applied Science and Engineering, College of Science and Engineering, Fu Jen Catholic University, New Taipei City 242, Taiwan; 10Brain Research Center, National Yang-Ming University, Taipei 112, Taiwan; 11Graduate Institute of Traditional Chinese Medicine, School of Traditional Chinese Medicine, Chang Gung University, Taoyuan 333, Taiwan; 12School of Nursing, National Taipei University of Nursing and Health Sciences, Taipei 110, Taiwan; 13Department of Traditional Chinese Medicine, Chang Gung Memorial Hospital, Keelung 204, Taiwan; 14Graduate Institute of Health Industry Technology, Chang Gung University of Science and Technology, Taoyuan 333, Taiwan; 15Department of Life Science, College of Science and Engineering, Fu Jen Catholic University, New Taipei City 242, Taiwan

**Keywords:** non-small-cell lung cancer, withaferin A, connectivity map, synergistic effect

## Abstract

Low response rate and recurrence are common issues in lung cancer; thus, identifying a potential compound for these patients is essential. Utilizing an in silico screening method, we identified withaferin A (WA), a cell-permeable steroidal lactone initially extracted from *Withania somnifera*, as a potential anti–lung cancer and anti–lung cancer stem-like cell (CSC) agent. First, we demonstrated that WA exhibited potent cytotoxicity in several lung cancer cells, as evidenced by low IC_50_ values. WA concurrently induced autophagy and apoptosis and the activation of reactive oxygen species (ROS), which plays an upstream role in mediating WA-elicited effects. The increase in p62 indicated that WA may modulate the autophagy flux followed by apoptosis. In vivo research also demonstrated the anti-tumor effect of WA treatment. We subsequently demonstrated that WA could inhibit the growth of lung CSCs, decrease side population cells, and inhibit lung cancer spheroid-forming capacity, at least through downregulation of mTOR/STAT3 signaling. Furthermore, the combination of WA and chemotherapeutic drugs, including cisplatin and pemetrexed, exerted synergistic effects on the inhibition of epidermal growth factor receptor (EGFR) wild-type lung cancer cell viability. In addition, WA can further enhance the cytotoxic effect of cisplatin in lung CSCs. Therefore, WA alone or in combination with standard chemotherapy is a potential treatment option for EGFR wild-type lung cancer and may decrease the occurrence of cisplatin resistance by inhibiting lung CSCs.

## 1. Introduction

Lung cancer is the leading cause of cancer death worldwide. According to the histological types published by the World Health Organization, lung cancer is classified into two basic types: small-cell lung cancer and non-small-cell lung cancer (NSCLC). NSCLC accounts for approximately 85% of all lung cancer cases, with 30% being squamous cell carcinoma (SCC) and the remaining (~70%), such as adenocarcinoma and large cell carcinoma, being collectively classified as non-SCC [1]. Despite efforts made for its early detection, approximately two-thirds of NSCLC cases are diagnosed at advanced stages with limited surgery options. For advanced NSCLC, the discovery of oncogenic driver mutations, such as epidermal growth factor receptor (EGFR) mutations and anaplastic lymphoma kinase gene rearrangements, has led to novel strategies of classifying NSCLC and the option of utilizing appropriate tyrosine kinase inhibitors (TKIs) for anticancer therapy. Although targeted therapy provides novel treatment options, chemotherapy remains the standard of care for most patients with NSCLC without driver mutations. Platinum-based chemotherapy is the treatment of choice for patients with adenocarcinoma and SCC [2]. However, the response rate of platinum-based chemotherapy is low, ranging from 20% to 40% [3]. In addition, progression-free survival (PFS) for platinum-based chemotherapy alone is only approximately six months. Several recent trials have demonstrated that immunotherapy combined with platinum-based chemotherapy may prolong PFS [4,5]. The benefit for overall survival remains unclear. Therefore, novel treatment modalities are still warranted.

The cancer stem cell (CSC) hypothesis states that malignant tumors mainly comprise differentiated cancer cells and are maintained by a small subpopulation of cells that give rise to phenotypically diverse cancer cells [6,7]. Studies have demonstrated that CSCs exist in various cancer types, and are responsible for tumor resistance to conventional therapeutics [8,9]. Therefore, targeting CSCs represents a potential strategy to improve the treatment outcome of cancer. After analyzing data from established studies, we noted that the expression of CSC-associated genes is correlated with poor clinical outcomes in acute myeloid leukemia [10], colorectal cancer [11], and non-small-cell lung cancer [12], as well as with drug resistance [13] and tumorigenesis [14]. The overexpression of ATP-binding cassette (ABC) transporters commonly observed in CSCs could facilitate their isolation through flow cytometry on the basis of their ability to pump out a fluorescent DNA binding dye, such as Hoechst 33342, representing the so-called side population (SP) cells [15]. The tumor spheroid-forming ability also represents a key stem cell-like property [16]; some markers, such as CD44, CD133, Oct-4, or c-Myc, are also overexpressed in CSCs [14,17]. The signaling pathways related to stemness play a major role in CSC-related phenotypes; for example, the activation of STAT3 and Notch signaling increase sphere formation, aldehyde dehydrogenase+ (ALDH) cell population, and stemness marker expression [18].

Because lung CSCs play a crucial role in sustaining the malignant phenotype of lung cancer and may cause resistance to conventional chemotherapy, we applied an in silico drug identification method to identify potential drugs that may reverse carcinogenic and CSC-associated gene signatures. Through vigorous elimination processes, we identified withaferin A (WA) as a potential candidate. WA is an isolate from the plant *Withania somnifera*, which has been used to treat a variety of conditions owing to its anti-inflammatory and antibacterial properties. WA has been reported to have anticancer [19], anti-angiogenic [20], and pro-apoptotic properties [21] in various cell types. In this study, we demonstrated that WA may induce cell apoptosis and autophagy through a reactive oxygen species (ROS)-dependent pathway. In addition, STAT3 signaling was downregulated in NSCLC stem-like sphere cells after WA treatment. The combination of WA and cisplatin and other chemotherapy drugs enhanced cytotoxicity for NSCLC cells. An in vivo study also revealed that WA inhibited NSCLC tumorigenesis in NOD/SCID mice. In conclusion, through in silico drug screening and multiple approaches for targeting CSCs, we identified WA as a potential anticancer candidate for NSCLC that may also be able to overcome cisplatin resistance by targeting CSCs.

## 2. Results

### 2.1. In Silico Screening Identifies WA as a Potential Candidate for the Treatment of NSCLC

The idea of drug repurposing—that is, finding novel therapeutic applications (i.e., new diseases) for known drugs and compounds—has been increasingly demonstrated in several studies [22,23,24]. In this study, we attempted to apply the CMap technique to identify drugs that facilitate a negative correlation between drug perturbation and lung cancer gene expression patterns, and which may therefore be potential drugs for treating lung cancer. Using gene signatures from three lung cancer array cohorts that prioritized three gene selection methods, we searched the CMap database and identified a total of 308 potential drugs that could significantly reverse lung cancer expression patterns to adjacent nontumor gene expression patterns. Fifty-seven drugs were subsequently selected based on their coexistence in the three query methods for CMap (Figure 1A and Appendix A), and we screened their IC_50_ values for various NSCLC cells through the MTT (3-[4,5-Dimethylthiazol-2-yl]-2,5-diphenyltetrazolium bromide) assay. Our analysis revealed that several drugs were potential candidates, and an initial in vitro screening indicated a low IC_50_ (<1.5 μM). Among these drugs, emertine had the lowest IC_50_, but it is known to induce vomiting, which may have a negative impact on a patient’s quality of life. Tanespimycin and sanguinarine are highly toxic, whereas trichostatin A is mainly used as a tool compound in research. Among these drugs, we focused on WA in this study. WA demonstrated the most significant anticancer effect for lung cancer cells. We further used the SRB assay to examine the cytotoxic effect of WA in NSCLC cells, such as A549, CL141 and H441 (adenocarcinoma with wild-type EGFR), CL97 and H1975 (adenocarcinoma harboring EGFR T790M mutation), and CL152 (SCC) and H1299 (large cell carcinoma). As illustrated in Figure 1B and Appendix A, NSCLC cells were sensitive to WA treatment, as reflected by their low IC_50_ values.

### 2.2. WA May Perturb Autophagy Flux and Induce Apoptosis in NSCLC Cells

The antiproliferative effect of WA was in part due to the induction of apoptosis, as WA treatment for 24 h caused the cleavage of caspase 3 in various lung cancer cells in a dose-dependent manner (Figure 2A). Several mechanisms, such as ROS generation, have been linked to WA-mediated anticancer effects [25]. To verify the effect of WA on ROS, live-cell imaging was performed to visualize ROS signal distribution and intensity according to distinct durations of WA treatment. ROS signals in H1975 cells were weakly detected in the control group and increased shortly after treatment with WA, suggesting that the increased ROS level was one of the early events caused by WA. The effect was prolonged after 24-h treatment with WA and was sufficiently blocked by 30 min pretreatment with *N*-acetyl-L-cysteine (NAC), a general ROS scavenger (Figure 2B). Considering that ROS production is also implicated in the induction of autophagy, we subsequently examined whether WA could induce ROS and autophagy in these proapoptotic cells. The time point experiment revealed that WA induced autophagy after 4 h of treatment and subsequently induced apoptosis after 24 h of treatment in H1975 cells. The increasing cleavage of caspase 3 and caspase 8 and the decrease in Bcl2 and Beclin-1 indicated that WA effects on H1975 cells shifted from autophagy to apoptosis at 24 h (Figure 2A,C). In addition, ROS plays an upstream role in WA-elicited autophagy and apoptosis, as indicated by the co-treatment with NAC, which significantly decreased WA-induced acridine orange (+) cells and Annexin-V (+) cells in H441, H1975, and CL152 cells (Figure 2D,E). Similarly, cotreatment with NAC rescued WA-induced cytotoxicity in H441, H1975, and CL152 cells (Figure 2F). Again, we demonstrated that WA induced the expression of heme oxygenase-1 (HO-1), which can be induced in response to oxidative stress, and increased the LC3-II/LC3-I ratio as well as cleaved caspase 3 (c-caspase 3) inhibited by the co-treatment with NAC (Figure 2G). Notably, although the LC3-II/LC3-I ratio increased after WA treatment, the p62 level increased, rather than decreased, by autophagy. This observation indicated that WA may not only induce autophagy but also affect the autophagy flux. Chloroquine, an inhibitor that blocks the fusion of lysosomes and autophagosomes, slightly increased p62 levels. Cotreatment with chloroquine and WA showed similarly increased p62 levels and the cleaved caspase 3. The aforementioned results imply that WA may play a role in the disturbance of the autophagy flux in a ROS-dependent manner, which subsequently induced apoptosis in lung cancer cells.

Nuclear factor E2-related 2 (NRF2), which plays an important role in antioxidant defense in normal cells, has been suggested to be activated in many types of cancer, such as lung cancer [26]. By disrupting the interaction with KEAP1-E3 ubiquitin ligase, accumulated and dysregulated NRF2 may contribute to tumor development and chemoresistance, suggesting that inhibiting NRF2 is a promising strategy for cancer therapeutics. Recently, the endogenous protein-protein interactions (PPIs) have been empirically detected using an in situ proximity ligation assay (PLA), which detects and visualizes endogenous PPIs with a high sensitivity and specificity. By utilizing Duolink PLA technology, we examined the KEAP1-NFR2 interaction as indicated by the presence of deep red blobs in cells. A reduction in the number of deep red blobs under 30 min WA treatment for H1975 cells indicated that WA could interrupt the interactions of NRF2-KEAP1, which might result from, at least in part, ROS and the subsequent autophagy mechanism. Although the interaction of KEAP1-NFR2 was decreased at early under WA treatment for 30 min, the interaction was increased upon WA treatment for 24 h (Figure 3A). Interestingly, we found WA treatment gradually increased KEAP1, while it decreased NRF2 in H1975 cells (Figure 3B), which correlates with the 24-h WA treatment in Figure 3A. These observations raised the possibility that WA may inhibit the cytoprotective abilities of cells via regulating the NRF2/KEAP1 pathway.

### 2.3. WA Inhibits Lung Tumorigenesis In Vivo

Subsequently, we used a lung tumor-bearing mouse model to validate the anti-NSCLC effect of WA. H441-L2G cells (lung adenocarcinoma cells with wild-type EGFR, which express the L2G fusion construct containing dual optical reporters (firefly luciferase 2 and eGFP)) were subcutaneously injected into NOD/SCID mice for in vivo validation of WA-mediated anticancer effects. Tumor burden was monitored using the bioluminescent (BLI) imaging technique. Mice receiving 2 mg/kg WA exhibited a lower tumor burden, which was reflected by the lower bioluminescence intensity compared with that of control mice (Figure 4A). Semiquantitative analysis indicated that WA treatment significantly suppressed H441 tumorigenesis, as was apparent in bioluminescence images (Appendix A). WA-treated animals exhibited longer survival rates than their vehicle-treated control counterparts (Figure 4B). No differences in body weights were observed in mice treated with WA and control mice (Figure 4C). These findings suggest that WA treatment significantly suppressed lung tumorigenesis in vivo at the given dosage without eliciting apparent systemic toxicity in the animals.

### 2.4. WA May Act as a Potential Anti-CSCs Drug

Through gene set enrichment analysis (GSEA) analysis, we further demonstrated that the WA-induced gene signature had an inverse correlation with stemness-associated gene signatures (Figure 5A), implying that WA may also exhibit anti-CSC activity. CL141 cells were cultured in a selective medium for one week to promote the enrichment of lung CSCs, as previously reported [27]. We demonstrated that the floating spheres formed by CL141 cells were highly enriched by CSCs through the evaluation of the expression of stem cell markers such as CD133, ALDH1A1, and Nanog (Figure 5B). WA exhibited a prominent inhibitory effect on the viability of these CSCs, as reflected by the lower IC_50_ values (Figure 5C). The mTOR/STAT3 signaling pathway has been reported as a requirement for the growth of stem cell-like breast cancer SP cells [28], and it is implicated in the mediation of WA-induced apoptosis. In addition, WA has been reported to inhibit STAT3 and promote tumor cell death [29]. Therefore, we explored whether WA can downregulate the mTOR/STAT3 signaling pathway in lung spheroid cells. In this study, we noted that WA also played a role in inhibiting mTOR and STAT3 signaling, as indicated by the decrease in phospho-mTOR and p-STAT3 in CL141 sphere cells (Figure 5D).

We further used the SP method to determine the effects of WA on lung CSCs. H441, CL97, and H1975 cultures contained a Hoechst 33342 efflux SP representing 2.14%, 8.79%, and 7.41% of cells, respectively (Figure 5E,F). WA appeared to effectively suppress SP cells that represented CSCs. Given that sphere formation can be used to identify the stem cell characteristic of self-renewal in vitro, we also examined the efficacy of WA in lung cancer spheroid formation. Our results indicate that WA can inhibit spheroid-forming capacity in the spheroid subpopulation of H441 lung cancer cells (Figure 5G), suggesting that WA can inhibit the self-renewal of lung CSCs.

### 2.5. WA Synergistically Enhances Cisplatin-Mediated Cytotoxicity in Lung Cancer And Lung CSCs

Platinum-based chemotherapy is the first-line treatment for NSCLC with wild-type EGFR, but the response rate is unsatisfactory and recurrence occurs in most patients. Recent evidence suggests that CSCs cause drug resistance and can be enriched by chemotherapeutic drugs such as cisplatin [9]. Therefore, identifying drugs that target CSCs, or that can be combined with standard chemotherapy, may increase the therapeutic response and reduce the recurrence of NSCLC. We thus explored whether WA could synergistically enhance chemotherapy-mediated cytotoxic effects. Combination index (CI) analysis was performed and revealed that the combination of WA and cisplatin or pemetrexed exerted synergistic antiproliferative effects at almost all concentrations tested (Figure 6A and Table 1), as evidenced by CI values less than 1. By combining WA and cisplatin at a constant ratio (1:10), we validated the synergistic cytotoxic effect of WA and cisplatin (Figure 6B). We also performed immunoblotting to demonstrate that WA treatment can inhibit mTOR/STAT3 and facilitate apoptotic caspase 3 cleavage. Moreover, the administration of WA further sensitized cells to cisplatin-mediated cytotoxicity (Figure 6C, compare western blot lane 4, 5 vs. 6, 7).

Subsequently, we determined the anti-self-renewal ability mediated by WA using tumor spheroids generated from H441 cells. The number of tumor spheroids decreased with treatment using WA alone or in combination with cisplatin. Cisplatin alone appeared to be ineffective (Figure 7A). Subsequently, we examined WA effects on lung spheroids using another key stemness determinant, ALDH activity. Our results clearly demonstrated that the proportion of ALDH+ cells significantly decreased with WA treatment alone or with WA plus cisplatin. A significant change was also observed between the combination treatment and cisplatin-only treatment (Figure 7B). All data suggest that WA and cisplatin synergistically inhibit the growth of lung cancer and have better inhibition effects on lung CSCs, thus benefiting NSCLC treatment.

## 3. Discussion

Accumulating evidence indicates that lung cancer is an extremely heterogeneous disease, as reflected by the numerous gene mutations observed even in a single tumor sample [30]. This genetic complexity represents a major obstacle for developing effective anti–lung cancer agents. More importantly, this complexity could be the major underlying reason why targeted therapeutic agents eventually become ineffective [31,32]. Therefore, in a more macroscopic approach, we chose to identify effective antagonists against lung cancer through CMap, which is a platform containing arrays with drug-modulating genomic distribution changes. We selected candidate drugs with top-ranked scores for reversing lung cancer tumor to nontumor differential expression patterns. We hypothesized that the drugs with the highest frequency detected from various datasets after searching CMap would have the highest potential to inhibit lung tumorigenesis by shifting (either entirely or partially) a tumorigenic genomic profile toward a less or nonmalignant (normal cell-like) genome. In addition, through GSEA analysis to compare the stemness-associated gene signatures with WA-modulated gene signatures, a similar negative correlation was observed. Therefore, WA is a potential agent for inhibiting lung tumorigenesis, as well as an anti–lung CSC compound, which was established in silico and in vitro.

Anticancer activities associated with WA have been reported in various cancer types. For instance, WA was demonstrated to elicit oxidative stress (ROS), leading to mitochondrial dysfunction and subsequent programmed cell death in leukemia cells [33]. In breast cancer, WA induced apoptosis through the induction of Bim-S and Bim-L in estrogen-responsive MCF-7 cells and in triple-negative MDA-MB-231 cells [34]. In addition, WA was reported to exhibit antiangiogenesis activity by binding to the intermediate filaments vimentin and F-actin [35], as well as nestin, another filament protein that regulates the TGF-β1-induced epithelial–mesenchymal transition in pancreatic ductal adenocarcinoma [36]. In this study, we first identified WA as a candidate for reversing lung/lung CSC gene signatures through bioinformatics analysis. Subsequently, we demonstrated that WA induced apoptosis and suppressed self-renewal ability in a panel of lung cancer cell lines. Mechanistically, WA inhibited oncogenic signaling in lung cancer, namely, the mTOR/STAT3 pathway, indicating that WA exerted an anti–lung cancer effect. According to previous studies, STAT3 inhibition sensitizes BEZ235 (a dual inhibitor of PI3K and mTOR) and increases LY294002 (PI3K inhibitor)-induced cell death [37]. Recent studies have indicated that STAT3 signaling not only plays an essential role in lung tumorigenesis, but also in maintaining lung tumor-initiating cells [38,39,40]. These data support our observation that WA treatment suppresses cell viability in lung cancer cell lines and tumor sphere-forming ability by downregulating STAT3 signaling. In sum, this study reveals that WA is an effective agent for inhibiting CSC growth by affecting multiple targets of the mTOR/STAT3 pathway in NSCLC.

To provide further preclinical evidence for using WA as an anti–lung cancer agent, we also examined the possibility of using WA in a synergistic condition. Cisplatin is the standard chemotherapy drug for inoperable patients with lung cancer. When combined with other conventional chemotherapy drugs, an overall response rate up to 30% can be achieved [41], but this rate remains unsatisfactory. As presented by our data, WA suppressed cell viability in various lung cancer cell lines synergistically with cisplatin. In support of our data, Yao et al. demonstrated that the inhibition of STAT3 signaling could overcome cisplatin resistance, enhancing the effect of radiotherapy in vitro and in vivo [42]. The combination of STAT3 knockdown and gemcitabine can restore sensitivity to gemcitabine in pancreatic cancer cell line by inducing proapoptotic signals and increasing the cell population in G1 cell cycle arrest [43]. These studies have provided the ideal rationale and sound support for the usage of WA in combination therapy to overcome drug resistance. In clinical practice, platinum in combination with pemetrexed is the treatment of choice for patients with EGFR wild-type lung adenocarcinoma, whereas platinum combined with gemcitabine is the drug of choice for patients with lung SCC. However, both the response rate and PFS are poor. Our study demonstrated the synergistic effect of WA and cisplatin for A549 (adenocarcinoma), CL141 (adenocarcinoma), and H1299 (large cell carcinoma). WA also exhibited synergistic effects with pemetrexed for CL141 (adenocarcinoma) and H1299 (large cell carcinoma), as well as with gemcitabine for CL152 (SCC cells). These findings suggest that WA might be a novel and promising candidate for combination treatment in patients with EGFR wild-type NSCLC, either SCC or adenocarcinoma.

In agreement with our prediction from GSEA analysis (which showed WA may have an anti-CSCs effect), we found WA treatment can dose-dependently inhibit the slide population cells and spheroid numbers of different lung cancer cells (Figure 5E–G). In addition, combination of WA and cisplatin showed synergistic effect on inhibiting spheroid number in H441 cells (Figure 7A). Although we did not show the synergism of WA and cisplatin in inhibiting ALDH+ cells, the combination of WA and cisplatin indeed further decreased the number of ALDH+ cells. Considering that the combination of various drugs for cancer treatment may be tailor-adjusted to distinct cell lines, such as 1 µM cisplatin plus 0.5 µM WA in H441 cells (Figure 7A), the optimal concentration of the combination of cisplatin and WA needs to be further examined. Overall, our data indicate that WA has anti–lung cancer and anti–lung CSC effects, which can also improve the treatment efficiency of current lung cancer therapeutic drugs, such as cisplatin.

The role of autophagy in cancer cells is complex, because it has tumor-initiating and tumor-suppressing functions. The therapeutic outcomes of targeting autophagy are controversial, depending on the tumor types and treatment characteristics [44,45]. Autophagy is considered a useful strategy to induce cell death and suppress tumor growth in apoptosis-resistant tumors. Several drugs are already being used in clinics or in clinical trials to induce autophagy for the treatment of malignant tumors, such as mTOR inhibitor, vitamin D3 analog, and imatinib [45]. In the present study, although WA treatment induced autophagy, as indicated by the upregulation of LC3-II, the inhibition of the autolysosome function by chloroquine (CQ) could not reverse WA-induced apoptosis. Because of the increased expression of p62, rather than its decreased expression, WA-mediated cytotoxic effects observed under WA treatment may have partly resulted from the perturbation of autophagic flux rather than from autophagy induction.

Recently, NRF2/KEAP1 mutation or deficiency has been reported to increase tumor metastasis, resistance to oxidative stress in tumor cells [46,47], and recurrence in patients undergoing radiotherapy for lung SCC [48]. NRF2 transcription factor plays a dual role in signaling transduction. It has been reported to protect normal cells from oxidative stress [49] and to promote carcinogenic in NRF2-addicted cancer cells [50]. During autophagy, phosphorylated p62 competitively binds to KEAP1, thereby dissociating the NRF2 from KEAP1, which results in NRF2-dependent transcription and assistance in cell growth [51]. We examined the ROS-induced autophagy NRF2/KEAP1 pathway and discovered that WA disrupted NRF2-KEAP1 protein-protein interactions. Moreover, the PLA result displayed that long-term WA treatment increases the interactions of NRF2-KEAP1, stabilizes the KEAP1-NRF2 interactions and decreases the NRF2 level in H1975 cells. In contrast with previous studies on NRF2/KEAP1 inhibitory effects in normal cells, WA decreased the NRF2 level, which may further inhibit NRF2-induced chemoresistance. A previous study revealed that WA regulated the p-AMPK expression level in glioblastomas cells [52]; however, treatment with WA resulted in no significant changes in p-AMPK in the lung cancer cell lines tested (data not included). In sum, the effect of WA may vary in response to distinct cancer cells. Our characterization of WA in lung cancer cells may provide additional information for anticancer research.

## 4. Materials and Methods

### 4.1. Microarray Data Source

We analyzed a total of three cohorts of pairwise microarray datasets to identify the gene expression signatures distinguishing lung adenocarcinoma tumors from adjacent nontumor tissues. Dataset 1 included two lung adenocarcinoma microarray datasets generated in our laboratory. First, a total of 48 pairwise samples from 24 patients were hybridized to the Affymetrix HG-U133A platform, and an additional 50 pairwise samples from 25 patients were subjected to analysis with the HG-U133 plus 2.0 platforms. These two arrays have been deposited in the NCBI GEO and are accessible through GEO series accession numbers GSE7670 and GSE27262, respectively. Among these data are 70 stage-IA/IB and 28 stage-III/IV pairwise lung adenocarcinoma samples. Dataset 2 (GSE10072; EAGLE) was obtained from Landi et al. and included 48 stage-IA/IB/IIA/IIB (from 24 patients) and 18 stage-III/IV (from nine patients) lung adenocarcinoma pairwise samples that were analyzed using the Affymetrix HG-U133A chip. Dataset 3 (GSE19804; NTU) included 86 stage-IA/IB/IIA/IIB (from 43 patients) and 26 stage-III/IV (from 13 patients) lung adenocarcinoma pairwise samples that were analyzed using the Affymetrix HG-U133 plus 2.0 platform.

### 4.2. Gene Selection

In this study, we implemented three kinds of gene selection criteria, including top 100, volcano plot, and SAM, to generate up- and downregulated signatures for querying CMap. The aim of using three different methods of querying CMap was to compare the individualized tumor vs. normal profiles to all tumor vs. normal profiles. The first method identified 100 probesets, with the highest absolute ratios also showing a magnitude of change (two-fold) from our pairwise microarrays. Second, we used the volcano plot method [53], combining a two-sample paired *t*-test (*p*-value < 0.0001) and fold-change cut off to identify significant gene signatures. We used the leave-one-out method to split the dataset and generate gene signatures from each sub-dataset. *t*-test is a typical statistical analysis used to identify significant probesets. Third, significance analysis of microarrays (SAM), a statistical algorithm that can identify significant genes by specific two-sample paired *t*-tests [54], was used. SAM estimated the false discovery rate (FDR) value with one hundred permutations in this study. For gene selection, statistically significant differences were applied at the FDR of zero when the selected gene signature was less than 1000 probesets. On the other hand, the stemness gene signature was available from Wong’s study [55] and compared to the WA-modulated gene signature in CMap for GSEA analysis.

### 4.3. Cell Culture and Chemical

The NSCLC cancer cell lines A549, CL141, H441, CL97, H1975, CL152, and H1299 were maintained in RPMI medium supplemented with 10% fetal bovine serum (FBS, Gibco, Carlsbad, CA, USA), 2 mM L-glutamine, 100 U/mL penicillin, and 100 μg/mL streptomycin. Cells were maintained in a humidified incubator with 5% CO_2_ at 37 °C. Each dish contained 10 mL RPMI medium as the culture condition for NSCLC cells. WA was purchased from Sigma-Aldrich (St. Louis, MO, USA).

### 4.4. Cytotoxicity and Sulforhodamine B Assay

Cells were plated in 96-well plates at a density of 4000 cells per well in sextuplicate. The cells were treated on the second day with the indicated agents for 24 or 48 h. Cells were treated with different concentrations of WA, chemotherapy drugs (pemetrexed, cisplatin, or gemcitabine), or a combination of both agents. Cytotoxicity was assessed using the sulforhodamine B (SRB) assay [56]. Briefly, the medium was discarded, and the adherent cells were fixed by 100 μL of cold 10% trichloroacetic acid (w/v) in each well for 1 h at 4 °C. After fixation, cells were stained with 100 μL/well of 0.4% (w/v, in 1% acetic acid) SRB solution for 30 min at room temperature (RT) and then washed three times with 1% acetic acid. After air-drying, 100 μL of 10 mM Tris base was added to each well, and the absorbance was read at 540 nm. Cytotoxicity is expressed as the percent of cells in treated wells relative to the number of cells in the solvent-only control, which was set to 100%. Each experiment was performed independently at least three times in sextuplicate, and cytotoxicities are given as means ±SD.

### 4.5. ROS Detection in Living Cells

H1975 cells were seeded on 18-mm coverslips in a 12-well plate for 48 h before drug treatment. WA were added at a concentration of 2 µM for 30 h or 24 h. NAC was used as an ROS scavenger to suppress the ROS induction process, and pre-treatment with NAC 10 mM for 30 min was followed by a co-treatment with either WA or H_2_O_2_. H_2_O_2_ can highly stimulate the release of ROS, and thus was used as a positive control at a concentration of 0.1 mM and treated for 30 min. For the indication of ROS level in living cells, CellRox Deep Red (Thermo Fisher, Bend, OR, USA) was added in the culture medium with the working concentration 2.5 µM for 30 min, incubated simultaneously with WA or H_2_O_2_. After that, the medium was totally removed and coverslips were washed with PBS Ca^2+^ Mg^2+^. Each coverslip was assembled into a Ludin chamber containing CO_2_-independent medium supplemented with 10% fetal bovine serum (FBS, Gibco, Thermo Fisher), 100 U/mL penicillin, and 100 μg/mL streptomycin. The signal of CellRox Deep Red could be observed at a wavelength of 642 nm under a total internal reflection fluorescence (TIRF) microscopic system (iLas2, Roper, France). Images were taken by objective with a magnification of 100× (NA = 1.4). Average intensity of individual cells was analyzed by ImageJ, and a Student’s *t*-test was applied to compare the difference between the control and each treatment group, or between the treatment groups with and without NAC.

### 4.6. Flow Cytometry

To detect the autophagy and apoptosis cells, H441, H1975, and CL152 cells were plated in 6-cm dishes at a density of 300,000 cells per dish; tested drugs were added the day after seeding. Acridine orange staining was used to detect the autophagic cells, and lung cancer cells were stained with 1 μg/mL acridine orange (Sigma) for 15 min, trypsinized, and resuspended in cold PBS before detection. A FITC Annexin V apoptosis detection kit (BD Pharmingen, San Jose, CA, USA) was used to detect the apoptotic cells, and the experimental steps were followed cautiously, according to the protocol described. Unstained cells and cells stained with Annexin V or PI were used to gate the population of early and late apoptotic cells, respectively.

### 4.7. In Situ Proximity Ligation Assay

H1975 cells were seeded on 12-well plates containing 12-mm coverslips for 24 h before drug treatment. After 30 min or 24 h of drug treatment, the cells were washed twice with PBS Ca^2+^ Mg^2+^ and fixed with paraformaldehyde 4%, followed by PLA. The protocol of PLA was according to Chen and Huang [57]. Primary antibodies KEAP1 monoclonal antibody (1F10B6, Invitrogen) and NRF2 polyclonal antibody (ab31163, Abcam) were used at a dilution of 1:800. The secondary antibodies attaching probes were provided from the kit Duolink In Situ PLA Probe Anti-Rabbit MINUS (DUO92006, Sigma-Aldrich) and Probe Anti-Mouse PLUS (DUO92001, Sigma-Aldrich). The protein-protein interactions (PPIs) were detected with detection reagent FarRed (DUO92013, Sigma-Aldrich). Phalloidin-Alexa 488 nm and DAPI were used to visualize the cytoplasm and nucleus, respectively. Images were taken by using a spinning disk confocal microscopic system with a magnification of 60× objective. Total number of blobs was calculated as Z-projection via Metamorph software. A student’s *t*-test was applied to compare the difference between the control and each treatment group.

### 4.8. Side Population Assay

Lung cancer cells were treated with the tested drugs for 24 h. After trypsinization and cell counting, cells were incubated with 1 μg/mL Hoechst 33342 (Sigma, Mendota Heights, MN, USA) at 37 °C for 90 min either alone or in the presence of 100 μM verapamil (Sigma). An inhibitor of the ABC transporter was used to gate the SP in cancer cells. After 90-min incubation, cells were centrifuged immediately for 5 min at 1500 rpm and 4 °C and resuspended in ice-cold PBS. Cells were kept on ice to inhibit the efflux of the Hoechst dye, and 5 μg/mL propidium iodide (BD) was added to discriminate dead cells and viable cells. Finally, these cells were filtered through a 35-μm nylon mesh (BD) to obtain single-suspension cells. Cell dual-wavelength analysis was performed on a 4-laser CytoFLEX (Beckman Coulter, Brea, CA, USA). Hoechst 33342 was excited at 405 nm with a violet laser, and blue fluorescence was emitted with a 450/45 band-pass (BP) filter and red fluorescence with a 660/20 BP filter.

### 4.9. Spheroid Formation Assay

Single cells were plated in 24-well ultralow attachment plates (Corning Inc., New York, NY, USA) at a density of 5000 cells/mL in tumor spheroid culture medium, DMEM/F12 supplemented with 1% N_2_ Supplement (Gibco, Carlsbad, CA, USA), 10 ng/mL basic fibroblast growth factor (Peprotech), 10 ng/mL epidermal growth factor (Peprotech, Rocky Hill, NJ, USA) with 1% amphotericin B, penicillin, and streptomycin (Gibco) at 37 °C with 5% CO_2_. When passaged, tumor spheres were harvested, and spheroids were dissociated with TrypLE (Gibco). The sphere number was determined by counting the microscope image.

### 4.10. Western Blot Analysis

Cell samples were lysed in lysis buffer (50 mM Tris-HCl, pH 7.4, 5 mM MgCl_2_, 1% Nonidet P-40, 150 mM NaCl, 1 mM phenylmethylsulfonyl fluoride). Total protein was isolated and subjected to SDS polyacrylamide gel electrophoresis and electrotransferred onto PVDF membranes (Millipore). The protein detection was performed with the enhanced chemiluminescence (ECL, Millipore) method and was captured by a Luminescence Imaging System (LAS-4000, Fuji Photo Film Co., Ltd., GE Healthcare, Taiwan).

### 4.11. Combined Drug Analysis

Drug dilutions and combinations were made in RPMI-1640 medium immediately before use. Following drug addition, the 96-well plates were incubated for 48 h, and the SRB assay was performed to determine cell viability. Analysis of synergism between different agents in inducing cytotoxic death of cells was performed by median dose-effect analysis and calculation of combination index (CI) using the commercially available software Chou and Talalay (CompuSyn software) [58,59]. According to the recommendations of this methodology, the value of CI is represented as following: CI values less than and greater than 1 indicate synergism and antagonism, respectively, whereas a value of 1 indicates addition.

### 4.12. In Vivo Monitoring of WA-Mediated Anti–Lung Cancer Effects

Female NOD/SCID mice (four to six weeks of age) were purchased from BioLASCO, Taiwan. All protocols were approved by the institutional animal care committee of Taipei Medical University Hospital. NOD/SCID mice were first inoculated with luciferase-expressing NSCLC H441L2G cells intravenously (1 × 10^6^ cells/100 μL PBS, *n* = 4 per group). The dual optical reporter L2G fusion construct (firefly luciferase 2 and eGFP) was a generous gift from Dr. Sanjiv Sam Gambhir, Stanford University. H441-bearing mice were imaged weekly for tumorigenesis. Mice were injected with 100 μL of D-luciferin (i.p. 30 mg/mL, Biosynth) and anesthetized with isoflurane (2% in 1 L/min oxygen). The bioluminescence images were acquired using the IVIS 200 system (Caliper Life Sciences, Mountain View, CA, USA). Acquisition times ranged from 10 s (for later time points) to 2 min (for early time points). Data are expressed as fold change in bioluminescence (total photon flux weekn/total photon flux week1) and analyzed using Living Image 1.0 software (Caliper Life Science, Mountain View, CA, USA). The body weight of the animals was also monitored on a weekly basis.

### 4.13. Statistical Analyses

The statistical analysis was performed with a *t*-test to calculate the statistical significance, and *p* < 0.05 was considered significant. (* *p* < 0.05; ** *p* < 0.01; *** *p* < 0.001).

## 5. Conclusions

We demonstrated that WA effectively impedes cell growth in NSCLC cells and cancer stem-like sphere cells. In contrast with previous studies that focused on the effect of WA on drug-resistant CSCs and the reduction of stemness markers, in the present research we not only studied the effect on lung CSCs but also demonstrated the synergistic effects of WA with pemetrexed, cisplatin, or gemcitabine on lung cancer cells. Our findings provide evidence for the clinical consideration of using WA in combination with conventional chemotherapy drugs to treat patients with EGFR wild-type lung cancer.

## Figures and Tables

**Figure 1 cancers-11-01003-f001:**
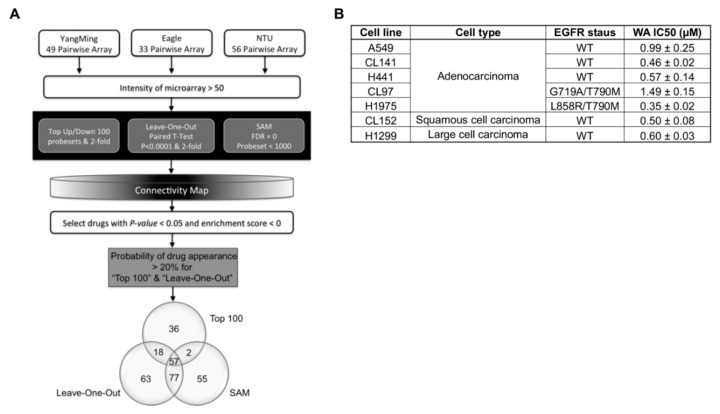
Identification of withaferin A (WA) as an anti–lung cancer agent using genomic approaches. (**A**) Yang Ming, Eagle, and NTU lung adenocarcinoma gene expression profiles were used for analysis. An intensity of microarray <50 implies a “cut value 50”, which indicates that the array intensities that are <50 after normalization will be rounded up to 50. To determine the effective criteria for selecting potential drugs, three methods were used to select gene signatures and create potential drug lists. Drugs appeared by top-ranked frequency when searching the CMap with significantly negative scores (*p* < 0.05, enrichment score <0) and were listed and compared with each other. Drugs with a frequency greater than 20% were selected based on three criteria. The results indicate that these compounds may potentially reverse the non-small-cell lung cancer (NSCLC) gene signature to the nonmalignant counterpart gene signature. (**B**) Of the 57 intersected compounds, WA was selected to test the IC_50_ at 48 h in NSCSC cells (*n* = 3). The epidermal growth factor receptor (EGFR) mutation statuses of these cells were also included. SAM: significance analysis of microarrays; WT=wildtype.

**Figure 2 cancers-11-01003-f002:**
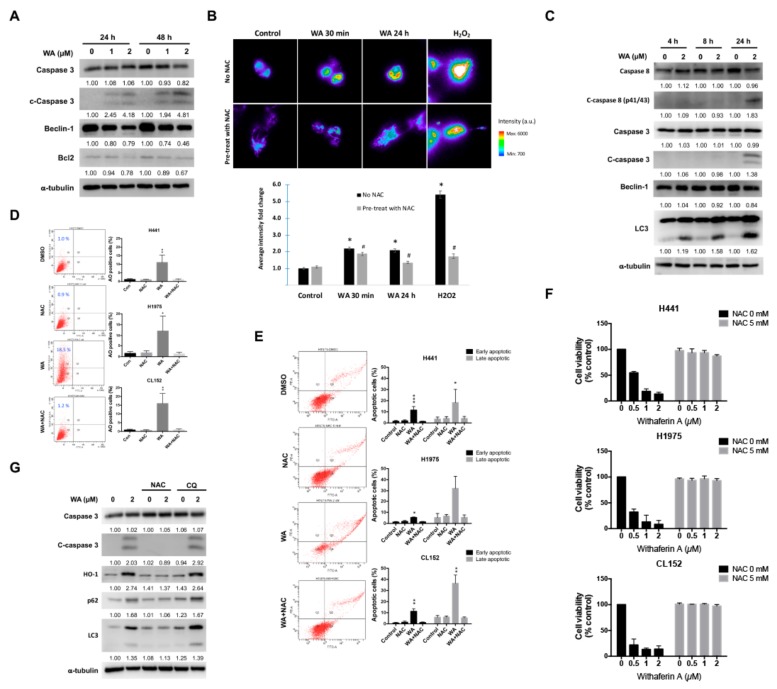
WA treatment induces autophagy in lung cancer cell lines. (**A**) H1975 cells treated with WA at indicated concentrations for 24 h or 48 h. Western blot analysis revealed an increase in the cleavage of caspase 3 (*n* = 3). (**B**) (Above) Representative images of ROS levels in various treatment groups. H1975 cells treated with WA at a concentration of 2 µM for 30 min or 24 h. A strong ROS inducer, H_2_O_2_, was used as a positive control and compared with WA. (Below) Quantitative analysis of the average fluorescence intensity presented as a fold-change (mean ± SEM) compared with the vehicle treatment group (DMSO). Approximately 30 cells were analyzed for each treatment group in three independent experiments * *p* < 0.05 vs. control; # *p* < 0.05 drug treatment with vs. without NAC (*N*-acetyl-L-cysteine). (**C**) WA induced autophagy in H1975 cells in a time-dependent manner, from 4–24 h after 2-µM WA treatment (*n* = 3). (**D**) (Left) Representative images of acridine orange staining in H1975 cells treated with DMSO, 2 μM WA, 5 mM NAC, and a combination of WA and NAC for 24 h. (Right) Quantitative analysis of acridine orange staining flow cytometry results from three lung cancer cell lines: H441, H1975, and CL152 (*n* = 3). (**E**) (Left) Representative images of PI-Annexin-V staining in H1975 cells treated with DMSO, 2 μM WA, 5 mM NAC, and a combination of WA and NAC for 48 h. (Right) Quantitative analysis of PI-Annexin-V staining flow cytometry results from three lung cancer cell lines: H441, H1975, and CL152 (*n* = 3). (**F**) Cell viability results of CL141, H441, H1975, and CL152 treated with WA (at 0.5, 1, and 2 µM) with or without 5 mM NAC (*n* = 3). (**G**) NAC suppressed WA-induced autophagy and apoptosis activation as indicated by the western blot analysis of H1975 cells (*n* = 3).

**Figure 3 cancers-11-01003-f003:**
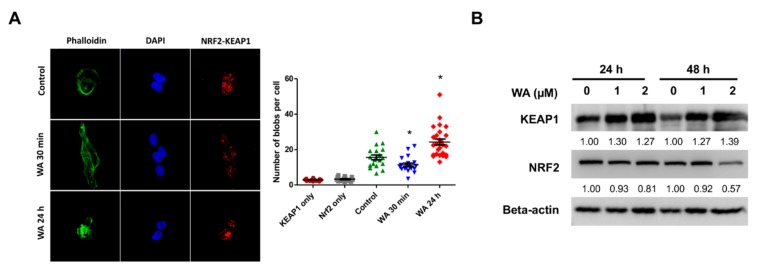
WA interrupts NRF2-KEAP1 interaction in NSCLC cells. (**A**) H1975 cells were treated with 2 μM WA for 30 min or 24 h. NRF2 and KEAP1 interactions were detected using the Duolink proximity ligation assay (PLA) kit. (Left) Representative images of each treatment with the deep red blob signal; NRF2-KEAP1 interactions were detected in the cytoplasm of H1975 cells. (Right) Quantitative analysis of PLA results; the blob number is presented as blobs/cell. * *p* < 0.05 vs. control. (**B**) WA treatment upregulated KEAP1 and downregulated NRF2 in H1975 cells (*n* = 3).

**Figure 4 cancers-11-01003-f004:**
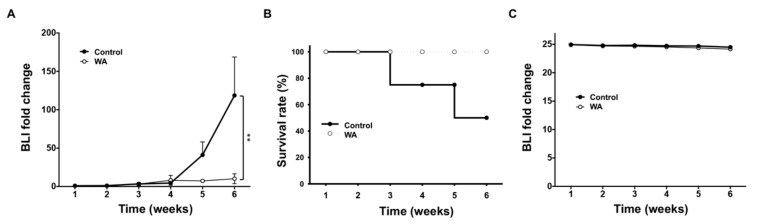
Noninvasive imaging of WA-mediated anti-lung tumorigenesis in vivo. (**A**) Semiquantitative analysis of tumor burden with and without WA treatment. Fold changes in bioluminescence (BLI) from both the control and WA groups were determined. WA-treated mice exhibited a significantly lower change in bioluminescence over six weeks when compared with control mice. ** *p* < 0.01 vs. control. (**B**) Kaplan–Meier survival curves indicated that mice receiving WA treatment had a better survival rate than controls over the six-week experimental period. (**C**) Longitudinal body weight monitoring revealed no significant difference between the control and WA-treated animals and suggesting no apparent systematic cytotoxicity at the dosage used in this study.

**Figure 5 cancers-11-01003-f005:**
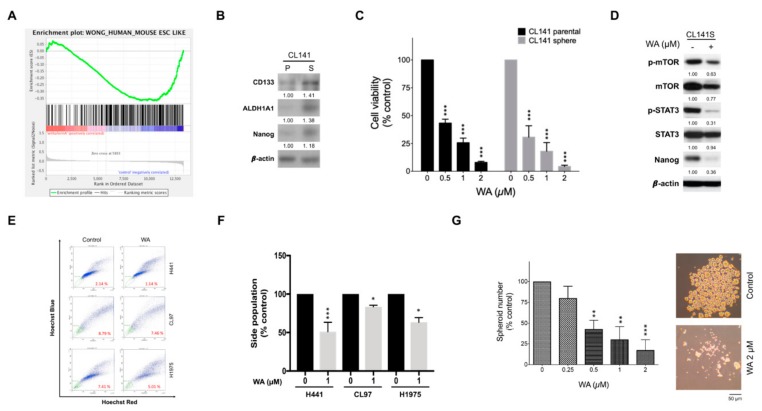
Identification of WA as an anti–lung cancer stem cell agent using GSEA analysis. (**A**) In GSEA analysis, WA drug signature appears to be the reverse of the embryonic stem cell module from Wong’s study, which is similar to the expression of CSCs. (**B**) In CL141 sphere cells, the protein expression of CD133, ALDH1A1, and Nanog was upregulated compared with the expression in their parental CL141 cells (*n* = 3). (**C**) WA treatment for 48 h significantly reduced the cell viability of CL141 sphere cells (*n* = 3). (**D**) CL141 sphere cells were treated with 0.5 µM WA for 24 h. Whole-cell lysates were resolved by SDS-PAGE and immunoblotted with specific antibodies, as indicated. WA inhibited mTOR and STAT3 activity, as indicated by the reduction of phospho-mTOR and STAT3 levels in CL141 sphere cells (*n* = 3). (**E**) H441, CL97, and H1975 cells were treated with 1 µM WA for 24 h, and the corresponding side population (SP) cells were gated after being stained with Hoechst 33342. (**F**) Quantitative analysis results indicate that WA decreased SP cells in lung cancer cells tested (*n* ≥ 2). (**G**) To analyze the effect of WA on sphere formation, H441 cells were cultured in stem cell medium before being treated with the indicated concentrations of WA for five days. The sphere number decreased in a dose-dependent manner after WA treatment (*n* = 4). ** *p* < 0.01, *** *p* < 0.001 vs. control.

**Figure 6 cancers-11-01003-f006:**
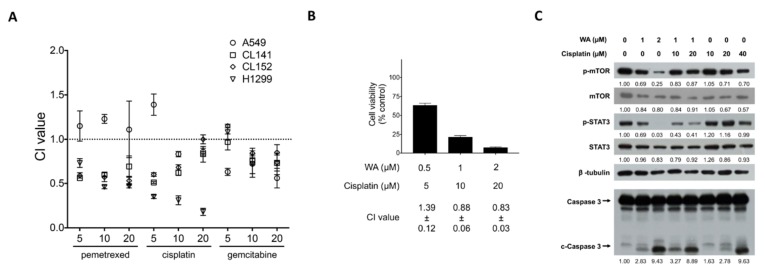
Synergy analysis of the interaction between WA and chemotherapy drugs in various lung cancer cell lines. (**A**) To analyze the synergistic effect of WA combined with chemotherapy drugs (pemetrexed, cisplatin, and gemcitabine), cells were exposed to 0.5 µM WA and 5, 10, or 20 µM chemotherapy drugs simultaneously for 48 h in A549, CL141, CL152, and H1299 cells (*n* = 3). (**B**) The cytotoxic effect and synergistic effect of WA combined with cisplatin were in constant ratio combination (1:10) on A549 cells for the 48-h treatment. The values of the CI were as follows: CI > 1, antagonism; CI = 1, additivity; CI < 1, synergism. The lowest CI value indicated the best synergistic effect of the combination of two drugs for the inhibition of cell viability. (**C**) H1975 cells were treated without or with WA and/or cisplatin as indicated for 24 h. Subsequently, cell lysates were harvested for the detection of caspase 3, pSTAT3/STAT3, p-mTOR/mTOR, and beta-tubulin by immunoblotting.

**Figure 7 cancers-11-01003-f007:**
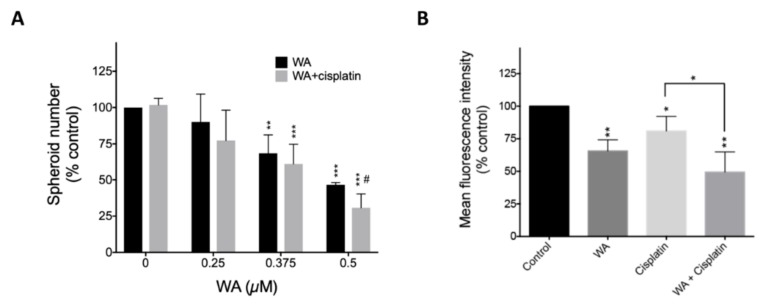
Combination of WA and cisplatin increases the ability of reducing cancer stem cell-related features as compare to cisplatin alone. (**A**) H441 sphere cells were treated without or with cisplatin (1 µM) and WA (0.25, 0.375, or 0.5 µM) as indicated for five days. Although cisplatin treatment did not reduce the sphere cell number, it further enhanced the cytotoxic effect of WA (at 0.375 and 0.5 µM). * *p* < 0.05; ** *p* < 0.01; *** *p* < 0.001 vs. control. In addition, cotreatment with WA (0.5 µM) and cisplatin had better effects compared with treatment with only 0.5 µM WA (*n* = 3). # *p* < 0.05 vs. corresponding dose of WA alone. (**B**) The mean fluorescence intensity of ALDH^+^ cells was tested after A549 cells were treated with 1 µM WA, 1 µM cisplatin, or their combination (*n* = 4). * *p* < 0.05 vs. control.

**Table 1 cancers-11-01003-t001:** The calculated combination index (CI) of synergistic interactions between withaferin A and chemotherapy drugs in NSCLC.

Cell Line	WA (µM)	(µM)	Pemetrexed CI	Cisplatin CI	Gemcitabine CI
A549	0.5	5	1.15 ± 0.17	1.39 ± 0.12	0.63 ± 0.04
		10	1.23 ± 0.05	0.83 ± 0.03	0.72 ± 0.15
		20	1.11 ± 0.32	0.86 ± 0.04	0.56 ± 0.11
CL141	0.5	5	0.56 ± 0.01	0.51 ± 0.01	0.97 ± 0.09
		10	0.60 ± 0.02	0.62 ± 0.03	0.76 ± 0.05
		20	0.69 ± 0.12	0.83 ± 0.09	0.73 ± 0.09
CL152	0.5	5	0.59 ± 0.03	0.60 ± 0.02	1.15 ± 0.02
		10	0.57 ± 0.05	0.68 ± 0.03	0.84 ± 0.06
		20	0.53 ± 0.05	1.00 ± 0.05	0.85 ± 0.09
H1299	0.5	5	0.73 ± 0.05	0.35 ± 0.02	1.08 ± 0.09
		10	0.46 ± 0.02	0.31 ± 0.05	0.72 ± 0.11
		20	0.47 ± 0.02	0.18 ± 0.04	0.72 ± 0.12

The calculated combination index (CI) determined by CompuSyn software. CI < 1.0 suggests a synergistic interaction between the two agents.

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
