# Peer review of "Identification of Withaferin A as a Potential Candidate for Anti-Cancer Therapy in Non-Small Cell Lung Cancer"

_cancers, 2019, doi:10.3390/cancers11071003_

Round 1
Reviewer 1 Report
The authors have addressed all of my concerns.
Reviewer 2 Report
Manuscript described important findings and is acceptable for publication.
This manuscript is a resubmission of an earlier submission. The following is a list of the peer review reports and author responses from that submission.
Round 1
Reviewer 1 Report
The manuscript cover most of the data already published by several investigators using different organ cancer cell lines such as breast, ovarian, prostate, endometrial, cervical etc. There is no novelty in the data presented and experiment performed.
Reviewer 2 Report
1. In this study, the authors have described the bioinformatics methods that were used to select withaferin A (WA) out of 57 identified drugs for testing against lung cancer. In fact, it is stated (line 103-104) that “Among these drugs, withaferin A (WA) showed the most significant anticancer effect for lung cancer”. However, the data supporting this observation is either missing or has not been included. Clearly, as per the Supplementary Table 1, several drugs show IC50 values <1 uM (Withaferin A, Trichostatin A, Tanespimycin, sanguinarine) and emetine (<0.1). Thus, the authors need to present a stronger rationale for selecting withaferin A out of the 57 compounds in the face of equivalent or even greater potencies.
2. The authors have used cleavage of caspase 3 and upregulation of P62 as indicators of WA induction of apoptosis and autophagy. Additionally, NAC was used to block these effects of WA, leading to the conclusion that WA disturbed autophagy in ROS-dependent manner. In this study, authors have not provided sufficient evidence to show that ROS production is required for the activity of WA and that ROS inhibition (by NAC) was related to decreased activity of WA. However, there is evidence in literature to show that inhibition of WA activity by NAC maybe related to the binding of WA to NAC rather than the ability of NAC to scavenge ROS. Further, ROS production has been shown to be secondary to the activity of WA. Therefore, the author’s conclusion in this regard may need modification since the data presented in the present study is insufficient to support the conclusion. For a detailed view, the authors can review the previously published study; https://www.ncbi.nlm.nih.gov/pubmed/25351115
3. In Fig 3 and Supplementary Fig 2, the authors show the in vivo efficacy of WA against NSCLC lung tumors in NOD/SCID mice. However;
a) There is discussion on the rationale for dose selection (2 mg/kg), dosing frequency, and duration are also not stated or discussed.b) The images in Supplementary Fig 2 (Week 1) show significant differences in the bioluminescence intensity (BLI) for control and WA groups. There is need to show that that the observed differences in BLI are related to treatment and not variability within experimental groups. Thus, the authors need to show that before treatments, there was sufficient randomization, and that average tumor burdens were similar in the control and intervention groups prior to the start of treatment. Otherwise, it can be concluded that the observed effects were not related to the treatment.
4. In Fig 5A, there is need to show dose-response data for cisplatin, permetrexed, gemcitabine, and WA, alone and in combination. Also, the each cell line can be plotted on separate graph and presented a panel within the figure. Fig 5C, the blots need quantification to tease out the actual contributions of each drug (WA or cisplatin) on mTOR and STAT3 activation.
5. In Fig 6, there appears to be no significant synergism owing to the overlap of error bars for the different treatment groups.
Reviewer 3 Report
Picture quality is very poor in figures.
Flow cytometry assay is not well described in material and methods. It is very important to know how CSC population has been defined. CSC population should characterize well. (Side population method)
Immunoblots must have quantification
Figure 6B doesn't show any significant difference in between cisplatin alone and combination.
Experiments related to autophagy induction with Wi-A are not clear. There are many pathways which regulate autophagy through ROS. Caspase activation with LC3I-LC3II upregulation doesn't make any correlation with anti-cancer properties of Wi-A. Authors should clarify which pathway is responsible for autophagy induction through ROS.
There is no information on direct target or other signaling molecules involved in Wi-A induced apoptosis. The targets are random and appeared in many other studies as well. the data lacks originality